# Mono- and Bilayer Graphene/Silicon Photodetectors Based on Optical Microcavities Formed by Metallic and Double Silicon-on-Insulator Reflectors: A Theoretical Investigation

**DOI:** 10.3390/mi14050906

**Published:** 2023-04-23

**Authors:** Teresa Crisci, Luigi Moretti, Mariano Gioffrè, Maurizio Casalino

**Affiliations:** 1Department of Mathematics and Physics, University of Campania “Luigi Vanvitelli”, Viale Abramo Lincoln, 5, 81100 Caserta, Italy; teresa.crisci@na.isasi.cnr.it (T.C.);; 2Institute of Applied Science and Intelligent Systems “Eduardo Caianiello” (CNR), Via P. Castellino n. 141, 80131 Naples, Italy; mariano.gioffre@na.isasi.cnr.it

**Keywords:** resonant cavity, photodetectors, near-infrared, silicon, graphene

## Abstract

In this work, we theoretically investigate a graphene/silicon Schottky photodetector operating at 1550 nm whose performance is enhanced by interference phenomena occurring inside an innovative Fabry–Pèrot optical microcavity. The structure consists of a hydrogenated amorphous silicon/graphene/crystalline silicon three-layer realized on the top of a double silicon-on-insulator substrate working as a high-reflectivity input mirror. The detection mechanism is based on the internal photoemission effect, and the light-matter interaction is maximized through the concept of confined mode, exploited by embedding the absorbing layer within the photonic structure. The novelty lies in the use of a thick layer of gold as an output reflector. The combination of the amorphous silicon and the metallic mirror is conceived to strongly simplify the manufacturing process by using standard microelectronic technology. Configurations based on both monolayer and bilayer graphene are investigated to optimize the structure in terms of responsivity, bandwidth, and noise-equivalent power. The theoretical results are discussed and compared with the state-of-the-art of similar devices.

## 1. Introduction

In recent years, the silicon (Si) photonics market has grown more and more promising to reach a value of USD 4B at the transceiver level [1]. The challenge of making Si suitable not only for electronics but also for sensors and photonics [2,3,4] has long attracted the interest not only of the scientific community but also of companies. At present, Si photonics is considered one of the most appealing approaches to overcome the criticalities of data centers exacerbated by the enormous increase in data traffic achieved in recent years, responding to the needs of low-power consumption, reliability, and low cost.

In this context, photodetectors (PDs) are key elements, as they allow the conversion of an optical information signal into an electrical signal. Unfortunately, the optical signals used for communications fall in the near-infrared (NIR) spectrum, precisely where the absorption of Si can be considered null or negligible due to the width of its bandgap.

Even if the use of germanium (Ge) is still a valid possibility [5,6], the difference in lattice constant with Si (4.3%) requires the realization of an adaptation buffer layer manufactured at high temperature [6,7,8,9] which, if on the one hand reduces the leakage current, on the other inhibits the monolithic integration of these PDs with electronics.

This creates the necessity to explore solutions for sub-bandgap photodetection in Si. In recent years, the internal photoemission effect (IPE) has gained some visibility in this field. It is based on the use of a Schottky junction, and the only requirement is that the incident radiation must have an energy greater than the potential barrier formed at the metal/semiconductor junction. Then, photoexcited carriers in the metal can be injected into the silicon, giving rise to a photocurrent and thus achieving the sub-bandgap photodetection into the silicon [10,11]. Cryogenic PDs based on IPE have already demonstrated the capability to detect the spectrum ranging from 1 to 5 μm [12,13,14,15,16], while in 2006, it was proposed to employ IPE also for NIR PDs operating at room temperature. Unfortunately, IPE is a very weak effect that leads to low-efficiency PDs. Many strategies have been employed to increase IPE, and among them are the use of optical microcavities [15], Si nanoparticles [17], surface plasmon polaritons (SPPs) [18,19], antennas [20], and gratings [21]. Despite this, PDs based on metal/Si Schottky junctions have shown limited performance characterized by responsivities below 30 mA/W [17,22]. These limited results mainly depend on the low probability of transferring photo-excited electrons from the metal to the Si. Recently, it has been demonstrated that such probability can increase as the metal becomes thinner [23,24]: the idea to replace metal with the 2-dimensional graphene (Gr) layer arises from these considerations. Schottky Gr/Si PDs based on IPE have shown unexpected performance in the infrared spectra [25], but the low absorption of graphene (2.3%) strongly compromises their efficiencies. In this regard, an advantageous option is the use of multiple reflections inside a resonant optical microcavity that can lead up to the full absorption of the incoming light. This approach has been already reported in the literature. However, the difficulty of realizing a high-finesse optical microcavity in silicon has impeded the realization of high-responsivity graphene/Si photodetectors. In fact, monolayer graphene transferred to the edge of a Fabry–Pèrot optical microcavity has provided a graphene absorption of 8% and a responsivity of 20 mA/W [26], while graphene incorporated between amorphous and crystalline Si has demonstrated an increase in the optical absorption of the 12% and responsivity of 25 mA/W [27]. Finally, a Fabry–Pèrot microcavity based on III-V semiconductors has shown a graphene optical absorption of 60% but a limited responsivity of only 21 mA/W and, of course, no compatibility with silicon technology [28].

In previous work, we proposed to realize a resonant-cavity-enhanced (RCE) photodetector based on a hydrogenated amorphous silicon (a-Si:H)/Gr/crystalline silicon (c-Si) microcavity built on top of a double silicon-on-insulator (DSOI) substrate, manufactured for high reflectivity at 1550 nm and working as an input mirror [29]. In this work, we showed that the employment of a distributed Bragg reflector (DBR), constituted by five periods of silicon nitride (Si_3_N_4_)/a-Si:H, as an output mirror was the best option to optimize the PD efficiency. On the other hand, the deposition of ten layers of Si3N4 and a-Si:H through a plasma-enhanced chemical vapor deposition (PECVD) system requires a high thermal budget, and makes the fabrication of these devices time-consuming. More importantly, the deposition of ten dielectric layers, constituting the high-reflectivity output DBR, causes stress accumulation on the 2-dimensional Gr layer and induces defects in the atomic lattice. Consequently, such a process may lead to the worsening of the Gr properties as well as the device performance.

In this work, with the main aim of simplifying the fabrication process of this kind of device, we propose to replace the a-Si:H/Si_3_N_4_ DBR output reflector with a thick gold (Au) layer. We have studied and optimized the new structure to combine significant technological simplification with high performance. The challenge of the proposed structure lies in the balance between the optical losses introduced by the gold, which can absorb the light without contributing to the photocurrent, and the optical absorption of the active layer of graphene. In our study, we considered both mono- and bilayer graphene, labeled as mGr and bGr, respectively. The numerical results are discussed and compared with those obtained for the device described in [29].

## 2. Figures of Merit for Photodetector Performance

Here, the most important parameters for the quantification of the PD performance are introduced.

It is well-known that the responsivity *R* is an important figure of merit for a PD, defined as the ratio between the photogenerated current (*I_ph_*) and the incident optical power (*P_inc_*). With regards to Schottky graphene/c-Si PDs, the responsivity, expressed in A/W [30], can be written as follows [31]:(1)$$R=\frac{I_{ph}}{P_{inc}}=A\cdot \frac{q}{hc}\cdot \lambda \cdot \frac{{\left(h\nu \right)}^{2}-{\left(q\phi_{B}\right)}^{2}}{2{\left(h\nu \right)}^{2}}\;$$
where *A* is the graphene optical absorption (it is a dimensionless parameter being the ratio between the optical power absorbed by the active layer and the incident optical power on the PD), *λ* is the wavelength, *q* = 1.602 × 10^−19^ C is the electron charge, *h* = 6.626 × 10^−34^ J⋅s is the Planck constant, *c* = 3 × 10^14^ nm/s is the light speed, *hν* is the photon energy, and *qϕ_B_* is the Schottky barrier height of the Gr/c-Si Schottky junction [31].

As reported in [29], the bandwidth for the kind of PDs here studied can be estimated as follows [32,33]:(2)$$f_{3dB}=\frac{1}{2\pi \left(\frac{t}{v_{sat}}+\frac{\pi \varepsilon_{0}\varepsilon_{s}R_{L}}{W}r^{2}+\frac{1}{2\pi \delta \nu }\right)}\;$$
where *ɛ*_0_ and *ɛ_s_* are the vacuum and silicon permittivity, respectively, *v_sat_* is the carrier saturation velocity in Si, *t* is the maximum distance that the electrons must travel before being collected (this distance is considered completely depleted), *R_L_* is the load resistance, *W* is the depletion layer, *r* is the radius of the area of graphene in contact with Si (considered circular), and *δν* is the spectral width of the absorption peak at half maximum [34].

Finally, the noise equivalent power (*NEP*) represents the minimum optical power that can be recognized by a PD. *NEP* normalized to the square root of the device area can be written as follows [35]:(3)$$NEP=\frac{\sqrt{2q\left(A^{*}T^{2}e^{-\frac{q\phi_{B}}{kT}}\right)}}{R}\;$$
where *T* is the absolute temperature, *k* is the Boltzmann constant, *A** is the Richardson constant (32 A/cm^2^K^2^ for p-Si [28]), and *R* is the responsivity defined in Equation (1).

## 3. Device Concept and Details on Numerical Simulations

In canonical RCE PDs, the cavity is constituted by a PIN diode surrounded by high-reflectivity DBRs, and all the involved materials are III-V semiconductors [36]. The three parts of the PIN structure are usually made with slightly different stoichiometric coefficients to reduce reflections at the interfaces and to consider the stacks as an entire cavity. Recently, we have proposed a new kind of cavity, consisting of an a-Si:H/Gr/c-Si hybrid structure. Here, the Gr, acting as an absorbing layer, is placed between the other two materials. The a-Si is leveraged as a buffer layer because of its refractive indexes very similar to the c-Si, which in addition to the small thickness of the Gr, allows the reduction of the Fresnel reflections at the interface and, consequently, the consideration of the a-Si:H/Gr/c-Si three-layer structure as one unified optical cavity. The buffer layer has the important role to accommodate the localized optical field on the thin Gr layer where the photodetection mechanism takes place through the Gr/c-Si Schottky junction. The optical cavity is built starting from a DSOI substrate, which has also the function of an input mirror. Indeed, it is composed of two c-Si/SiO_2_ periods whose thicknesses are optimized to have a resonance at 1550 nm. The novel element in our device is the metallic reflector on the top of the a-Si:H layer: being deposited through thermal evaporation, the Au layer provides a huge technological simplification concerning the one studied in [29]. It is worth noting that the Au metal mirror is not in direct contact with c-Si where the charge transport takes place, so we do not expect a reduction of the charge lifetime which would improve the response rate at the expense of the responsivity [37].

However, whereas on the one hand, the use of a thick Au layer as an output mirror facilitates the manufacturing process while still ensuring high reflectivity, on the other hand, it introduces optical losses in the cavity, causing a worsening of the performance in term of responsivity, bandwidth, and NEP. Proper optimization of the cavity is required to achieve high performance, otherwise, the thick layer of Au, which has a much higher extinction coefficient (and consequently an absorption coefficient) than the mGr, would absorb most of the optical radiation trapped in the cavity but without contributing to the photocurrent. Thus, the presence of the metal reduces the Gr optical absorption A of Equation (1) leading to a responsivity reduction as well as a negative repercussion also on the NEP described by Equation (3). In addition, the increase of losses in the cavity produces a broadening of the spectral width of the absorption peak at half maximum (δν in Equation (2)) hindering the device bandwidth. To understand the impact of the Au as an output mirror and how the decrease in performance can be mitigated, the structure sketched in Figure 1 has been numerically investigated. By taking advantage of interference phenomena inside the cavity, it is possible to make the optical absorption of the Gr slightly higher than the optical absorption of the 200 nm-thick Au layer. To this aim, both the cases of a mGr and a bGr have been considered and studied. 

The numerical analysis of the proposed device has been carried out by implementing the Transfer Matrix Method (TMM) [38] in MATLAB. The TMM is a mathematical method well suited for the analysis of electromagnetic wave propagation through a multilayer medium. Indeed, due to the linearity of the electromagnetic laws, a photonic system can be studied by using matrixes. More in detail, a structure formed by a stack of various layers can be described through a system matrix, if in each layer the assumption of an infinitely extended slab of homogeneous material is valid. The main hypothesis is simply the continuity conditions for the electric field across the interface. In such a way, it is possible to determine the reflection and transmission characteristics of a multilayer structure together with the field components. Finally, the absorption through the stack can be calculated by using the expression of the Poynting vector as discussed more fully in [39]. In this work, the planarity of the structure to be investigated fits well with the assumptions required by the TMM, and to make the numerical simulations as accurate as possible the dispersion relations of all materials involved in the proposed structure, reported in Figure 2a,b, have been taken into account.

In the spectral range of interest, only Au and Gr have non-zero extinction coefficients *κ*, as shown in Figure 2b. All dispersion curves of Figure 2 are taken by Refs. [40,41,42], while the graphene refractive index *n_g_* can be written as follows [43,44]: (4)$$n_{g}=\sqrt{\varepsilon_{g}}=\sqrt{5.7+j\frac{G\lambda }{2d_{g}}}\;$$
where *ε_g_* is the graphene relative permittivity, $G=\frac{q^{2}}{2\varepsilon_{0}hc}=0.0073$ is the fine structure constant in SI base units [45], and *d_g_* = 0.335 nm is the thickness of single layer graphene. Equation (4) suggests that the refractive index of the bGr is the same as the mGr in the NIR range. The absorption A of a material depends on its thickness multiplied by the absorption coefficient *α*, defined as *α = (4π/λ)κ*. Thus, from these considerations, it is easy to verify that the optical absorption of the Gr grows together with the number of layers. Very interestingly, this leads to an enhanced optical absorption A as shown in Figure 3a, where the optical absorption of a suspended Gr for both mGr and bGr has been computed using the TMM approach. 

As expected, Figure 3a shows a uniform optical absorption of around 2.3% of the suspended mGr in a wide range of wavelengths, and this value almost doubles for suspended bGr.

In Figure 3b, the spectral reflectivity of a c-Si/SiO_2_ DSOI with thicknesses of 340 nm/270 nm and a 200 nm-thick Au mirror is reported: the DSOI and the Au reflectors are characterized by a reflectivity of 0.89 and 0.98 at 1550 nm, respectively. Figure 3b shows that considering the absence of optical transmission in a 200 nm-thick Au mirror, the optical absorption of Au is comparable to that of suspended mGr (2.3%), and consequently, in a non-optimized cavity the absorption of the metal mirror, which does not generate electric charges, risks considerably degrading the efficiency of the PD.

## 4. Theoretical Results and Discussion

The structure displayed in Figure 1 has been numerically analyzed: it is a three-layer a-Si:H/Gr/c-Si microcavity, with a 200 nm-thick Au layer as an output mirror and a dielectric DSOI input mirror. This study has been carried out firstly considering a mGr and then repeated with a bGr. Figure 4a,b show how the optical absorption of mGr and Au at 1550 nm varies by changing the thicknesses of c-Si and a-Si:H, i.e., the length of the cavity, while Figure 4c shows that the maximum graphene absorption of 0.43 at 1550 nm can be obtained in an optimized cavity consisting of 110.8 nm-thick and 85 nm-thick of c-Si and a-Si:H, respectively. By setting these thicknesses, the Au optical absorption results 0.42 at 1550 nm, leading to total optical absorption in the cavity of 0.85 as reported in Figure 4c. It is worth noting that by the employment of the cavity, the optical absorption of a mGr can be increased by more than an order of magnitude with respect to a standalone mGr whose optical absorption is commonly known to be only 0.023.

By using both the mGr optical absorption shown in Figure 4c and Equation (1) as well as by considering *qϕ_B_* = 0.45 eV [26] and *hν* = 0.8 eV (photon energy at 1550 nm), the device responsivity has been calculated and reported in Figure 4d, from which it emerges as a maximum value of 0.18 A/W at 1550 nm.

The bandwidth of the devices has been computed by evaluating the time constants discussed in Section 2. First, the carrier transit time, *τ_tr_*, has been obtained by considering the path *t* that an electron generated in the center of the mGr disk must travel before being collected (*τ_tr_* = *t/v_sat_*) by the Si ohmic contact. Under the assumption that the path *t* is completely depleted and approximately equal to the radius *r* (*t* ≃ *r*), and by taking *v_sat_* = 10^7^ cm/s [32], the red solid curve in Figure 5a can be derived. On the other hand, as reported in Equation (2), the constant time *τ_RC_* = (*πr*^2^*ε*_0_*ε_s_R_L_*)/*W* is completely determined by the knowledge of the depletion region *W* = *√*((2·*ε*_0_·*ε_s_*)/*qN_a_*)·*V_bi_*, which in turn depends on the built-in voltage *V_bi_* = *Φ_B_* − (*E_F_* − *E_V_*). By taking the p-type doping *N_a_* = 10^15^ cm^−3^, the difference between the Fermi level and the valence band far from the junction in Si is *E_F_* − *E_V_* = 0.254 V and, by considering a potential barrier *qϕ_B_* =0.45 eV [26], values of *V_bi_* = 0.197 V and W = 0.5 μm can be estimated. Furthermore, if a load resistance *R_L_* of 50 Ω is employed, the *τ_RC_* exhibits the behavior traced by the blue dashed line reported in Figure 5a. Then, the cavity photon lifetime *τ_ph_* = 1/2*πδν* can be completely determined by the full width at half maximum (FWHM) of the absorption peak (13.6 nm) shown in Figure 4c which corresponds to *δν* = 1698 GHz. Being independent of the geometry of the device, the constant time *τ_ph_* is only influenced by the cavity optical properties, resulting consequently independent of the mGr disk radius *r* as displayed by the green dotted line in Figure 5a. As a result, the total 3 dB roll-off frequency as a function of the mGr disk radius described by Equation (2) has been derived and reported in Figure 5a. Here, it is clear how the transit time limits the performance of the device in terms of bandwidth, making necessary the reduction of the Gr disk radius *r* below 15 μm to achieve bandwidth above 1 GHz.

Finally, in Figure 5b, the spectral NEP is reported and evaluated by combining Equation (3) with the results of Figure 4d. To this aim, the following values have been used: *qϕ_B_* = 0.45 eV [26], *T* = 300 K, *k* = 8.617 × 10^−5^ eV/K, *A** = 32 A/cm^2^K^2^. Figure 5b shows a minimum NEP of 0.86 $W/\mathrm{cm}\sqrt{\mathrm{Hz}}$ at 1550 nm.

So far, unfortunately, the proposed PD is characterized by lower responsivity and higher NEP than the counterpart based on DBR investigated in [29], even if it has comparable bandwidth. From the above considerations, it is expected that the use of a bGr (thickness of 0.67 nm), absorbing light about twice compared to the mGr, may increase the performance of the device. Then, we have repeated the reasoning used about the mGr thus far considering a bGr embedded in the a-Si:H/c-Si cavity.

It is worth noting that the Schottky barrier of a bGr/c-Si (P-type) junction is reported in the literature as high as *qϕ_B_* = 0.44 eV [46], very similar to that one of the mGr/c-Si junction investigated above.

Figure 6a,b show the results of the optical absorption of both bGr and Au at 1550 nm obtained by varying the thicknesses of c-Si and a-Si:H, while Figure 6c reveals a maximum bGr absorption of 0.65 at 1550 nm, obtained in an optimized cavity consisting of 112.2 nm-thick and 83.6 nm-thick layers of c-Si and a-Si:H, respectively. In this case, the Au optical absorption results in 0.32 at the wavelength of interest, 1550 nm. As a consequence, total optical absorption in the cavity close to one can be achieved as reported in Figure 6c. The comparison of the results in Figure 2a and Figure 6a, demonstrates again, as in the previous case of the mGr, that the employment of the cavity helps to enhance the optical absorption of a bGr by more than an order of magnitude.

After the cavity optimization, the bGr can absorb most of the radiation trapped in the cavity at the expense of Au as displayed in Figure 6c. This leads to an enhanced responsivity reported in Figure 6d. By taking advantage of both the bGr optical absorption reported in Figure 6c and Equation (1) and by considering both *qϕ_B_* = 0.44 eV [46] and *hν* = 0.8 eV (photon energy at 1550 nm), the device responsivity has been derived and plotted in Figure 6d, exhibiting a maximum peak of 0.27 A/W at 1550 nm.

Moving our attention to the bandwidth, it has been computed as discussed above but both taking care to use *qϕ_B_* = 0.44 eV [46] and calculating the relative cavity photon lifetime, *τ_ph_* = 1/2*πδν*, by the absorption peak of Figure 6c whose FWHM is 15.4 that converted in frequencies gives the value δν= 1923 GHz. This value is higher than both the PD based on mGr and the PD investigated in [29]. In other words, the device based on bGr is characterized by the lowest selectivity, and it can be ascribed to the increased losses in the cavity. Moreover, the time constants and the total 3 dB roll-off frequency, as a function of the bGr disk radius calculated by Equation (2), were evaluated and plotted in Figure 7a, which shows as the transit time limits the performance of the device in term of bandwidth. Therefore, also in this case, it would be necessary to reduce the Gr disk radius *r* below 15 μm to obtain bandwidth above 1 GHz.

For such a device, the spectral NEP has been evaluated by Equation (3) by taking advantage of the responsivity plotted in Figure 6d. For the NEP calculation, the following values have been used: *qϕ_B_* = 0.44 eV [46], *T* = 300 K, *k* = 8.617 × 10^−5^ eV/K, *A** = 32 A/cm^2^K^2^. Figure 7b shows a minimum NEP of 0.58 $W/\mathrm{cm}\sqrt{\mathrm{Hz}}$ at 1550 nm, lower than both the PD based on mGr and the PD studied in [25]. Of course, the decrease in NEP is strongly linked to the increase in responsivity as is clear by looking at Equation (3).

All results coming out from our simulations are summarized in Table 1, where also the numerical results of [29] are added for comparison.

The structure proposed in this work, based on a metallic output mirror, provides a huge technological streamlining compared to that one based on a dielectric output mirror (DBR) investigated in [29]. Indeed, a simple deposition of 200 nm-thick Au by thermal evaporation instead of the deposition of ten layers by ten PECVD processes for DBR manufacturing, makes the fabrication of these devices more rapid and straightforward to be executed. More important, the lack of stress accumulated during the DBR fabrication helps to preserve the graphene properties. 

Of course, there is a price to pay for this simplification, indeed the structure based on mGr shows comparable bandwidth but reduced responsivity and higher NEP with respect to that one investigated in [29]. This is mainly due to the presence of the Au layer which absorbs the optical radiation in the cavity without contributing to the photogenerated current. To increase the optical absorption of the active absorbing mGr concerning the not active Au reflector, a bGr has been proposed. The resonant cavity PD based on bGr shows the highest responsivity together with the lowest NEP as also summarized in Table 1. The responsivity of 0.27 A/W is a very interesting value, especially if we consider that such devices could be realized at the same time (monolithically) as electronic circuitry. It is worth noting that we limited our investigation to bilayer graphene without extending it to few-layer graphene because the device efficiency depends not only on the optical absorption but also on the Schottky barrier formed at the graphene/silicon interface (Equation (1)). While in the literature monolayer and bilayer graphene show a low Schottky barrier of 0.45 and 0.44 eV, respectively, it seems that the Schottky barrier is higher by increasing the number of layers. Indeed, in [28], the Schottky barrier between few-layer graphene (4/5 layers) and silicon has been quantified as 0.67 eV showing a limited maximum responsivity of 25 mA/W.

On the other hand, the lowest NEP of 0.58 nW/cm $\sqrt{\mathrm{Hz}}$ associated with the bGr configuration could be very useful for many applications, including free-space optical communications, where high sensitivity is required. Interestingly, the speed of the devices based on both the Au output mirror and dielectric output mirror (DBR) remains unchanged being the bandwidth limited by the carrier transit time, i.e., by the geometry of the device. This remains a limitation for this type of device where the collection of carriers occurs transversely to the direction of the incident light. Indeed, by considering a circular Gr active area with a radius *r* = 70 μm, all devices are characterized by a bandwidth of only 186 MHz which can increase up to 1 GHz if the radius reduces to 15 μm. This is the reason why they are probably not indicated for applications where high speed is the main requirement. Finally, the devices based on bGr, due to the increased losses in the cavity, are characterized by the widest FWHM; therefore, their employment is unsuitable for highly selective PDs.

## 5. Conclusions

The performance of graphene/silicon photodetectors based on optical microcavities formed by metallic and double silicon-on-insulator reflectors has been theoretically investigated in this work. The introduction of a metallic mirror instead of a distributed Bragg reflector makes the realization of these devices feasible in practice. However, the introduction of an absorbing mirror reduces the quality factor of the cavity hindering the performance of the devices. Here, proper optimization of the device has been carried out to overcome this drawback working on both monolayer and bilayer graphene. From this numerical study, the absorption of monolayer graphene in such a cavity has emerged to be 0.43, as high as the gold layer resulting in a responsivity of 0.18 A/W and a NEP of 0.86 nW/cm$\sqrt{\mathrm{Hz}}$. These outcomes demonstrate no improvement if compared with the counterpart based on DBR, already investigated in the literature. On the other hand, bilayer graphene offers the possibility to improve the device’s performance. We proved that the absorption of bilayer graphene, in the optimized cavity, is two times higher than the absorption of the gold mirror being 0.65 and 0.32, respectively. As a consequence, the proposed PD based on bilayer graphene is characterized by a responsivity of 0.27 A/W and a NEP of 0.58 nW/cm$\sqrt{\mathrm{Hz}}$ exhibiting thus higher and competitive performance. However, it is worth mentioning that the PDs based on monolayer graphene show a selectivity higher than that of the same devices based on bilayer graphene, being the FWHM-calculated as 13.6 nm and 15.4 nm, respectively. Unfortunately, concerning the bandwidth, the investigated devices show performance limited by the carrier transit time.

Considering the good responsivity achievable by our device and taking into account the limited bandwidth, we believe this work can open the way to photodetectors integrable within silicon-based photonic integrated circuits for power monitoring applications, where high-speed is not the main requirement, but high sensitivity (low NEP) and reasonable responsivity are mandatory. Such a kind of device may represent a concrete contribution to overcoming the limit of all-silicon-based photodetectors in the NIR range combining good performance with practical implementation thanks to the technological simplification here proposed and analyzed.

At this point, experimental results are desirable and mandatory for confirming the predictions of the numerical simulations.

## Figures and Tables

**Figure 1 micromachines-14-00906-f001:**
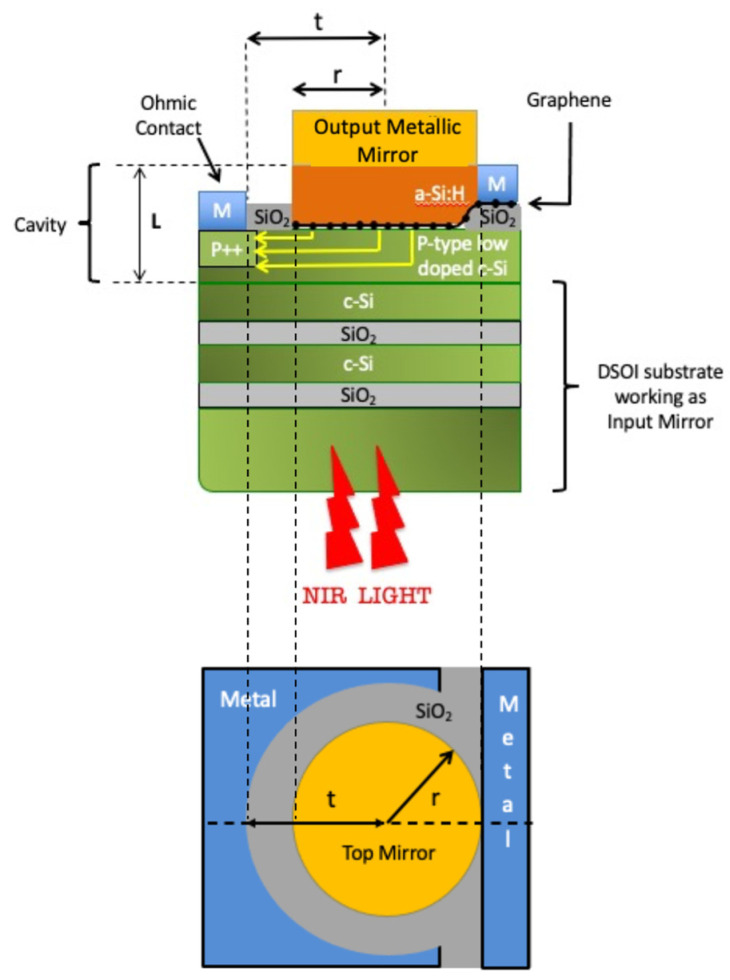
Sketch of the Metal/a-Si:H/Gr/Si/DSOI PD.

**Figure 2 micromachines-14-00906-f002:**
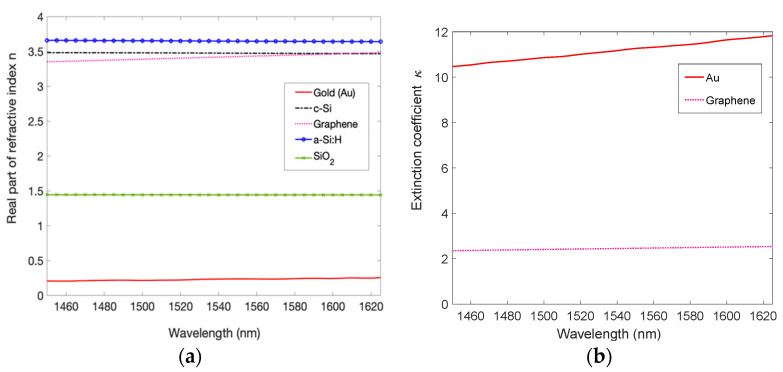
Dispersion curves of all materials used in our simulations: (**a**) real part of refractive index, (**b**) extinction coefficient.

**Figure 3 micromachines-14-00906-f003:**
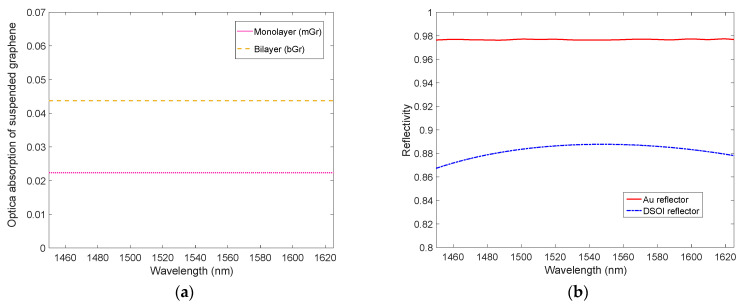
(**a**) Optical absorption of mono- and bilayer suspended graphene calculated by Equation (4) and (**b**) reflectivity vs wavelength of both the input mirror (DSOI) and the output mirror (200 nm-thick Au metal).

**Figure 4 micromachines-14-00906-f004:**
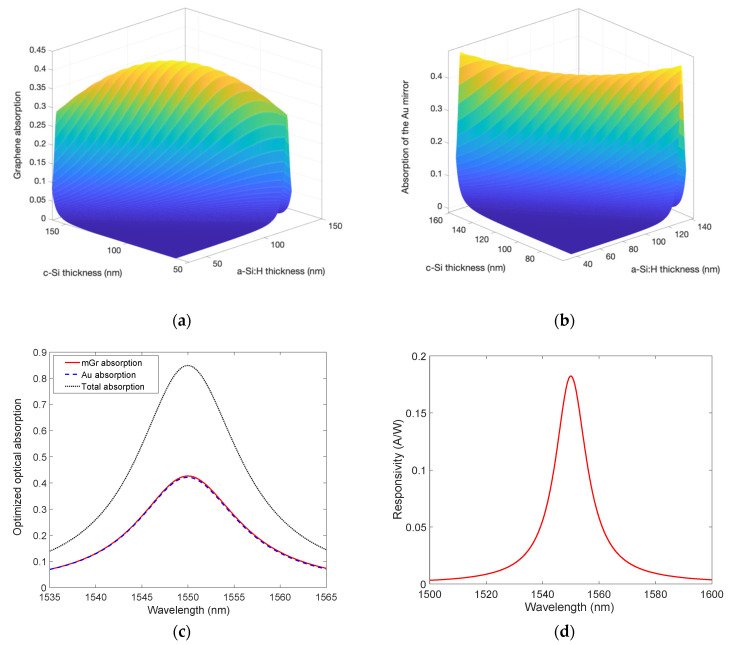
Optical absorption as a function of both c-Si and a-Si:H thicknesses of (**a**) mGr, (**b**) 200 nm-thick Au, (**c**) spectral optical absorption of mGr and Au and total absorption in the optimized cavity, and (**d**) spectral responsivity of the optimized mGr/Si Schottky PD.

**Figure 5 micromachines-14-00906-f005:**
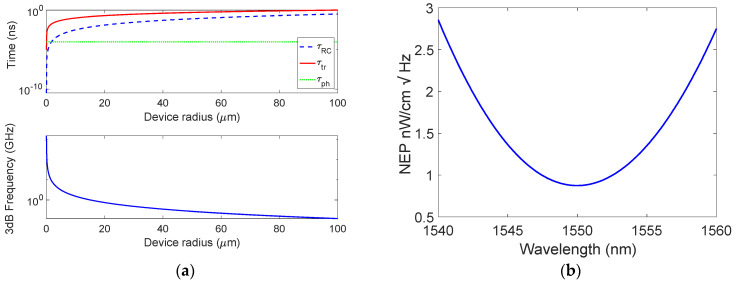
(**a**) Constant times characterizing the proposed PD and 3 dB roll-off frequency as a function of the mGr disk radius, and (**b**) spectral NEP.

**Figure 6 micromachines-14-00906-f006:**
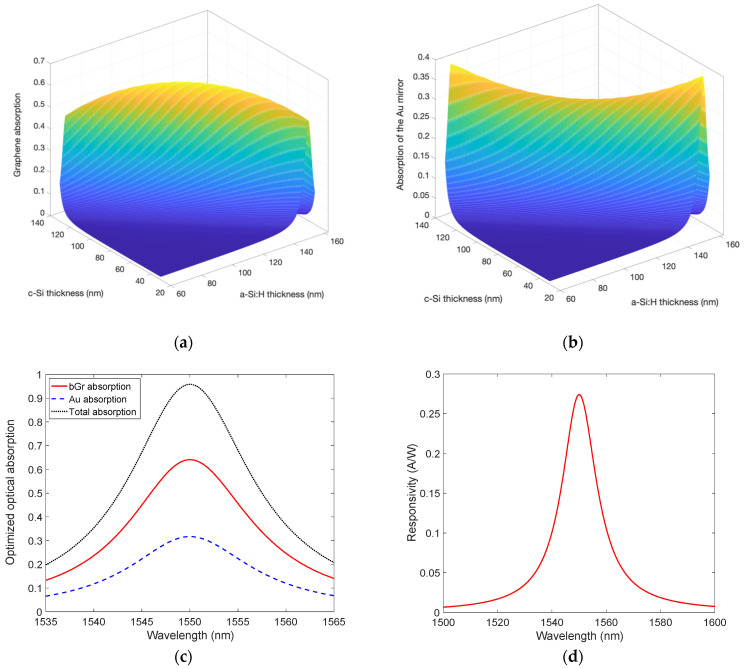
Optical absorption as a function of both c-Si and a-Si:H thicknesses of (**a**) bGr, (**b**) 200 nm-thick Au, (**c**) spectral optical absorption of bGr and Au and total absorption in the optimized cavity, and (**d**) spectral responsivity of the optimized bGr/Si Schottky PD.

**Figure 7 micromachines-14-00906-f007:**
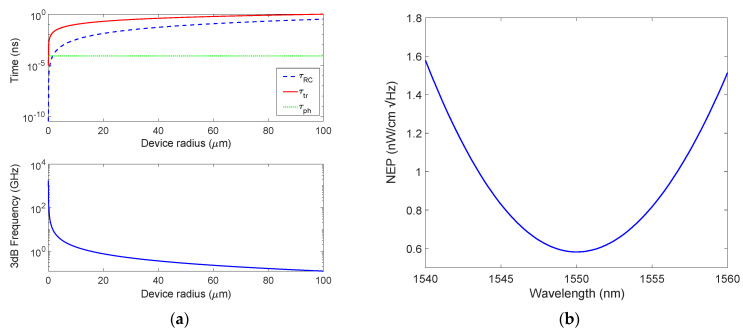
(**a**) Constant times characterizing the proposed PD and 3 dB roll-off frequency as a function of the bGr disk radius, and (**b**) spectral NEP.

**Table 1 micromachines-14-00906-t001:** Summary of the numerical results coming out from our simulations in comparison with numerical results reported in [29].

Input Mirror	Output Mirror	c-Si Thick. nm	a-Si:H Thick. nm	FWHM nm	Resp. @1550 nm A/W	3 dB Freq. @r = 70 μm MHz	Resp. × 3 dB Freq. A/W × MHz	NEP at 1550 nm nW/cm$\sqrt{\mathrm{Hz}}$
DSOI (mGr) [29]	DBR (5 pairs)	111.0	214.0	8.54	0.24	186	44.60	0.60
DSOI (mGr)	Metal	110.8	85.0	13.60	0.18	186	33.48	0.86
DSOI (bGr)	Metal	112.2	83.6	15.40	0.27	186	50.22	0.58

## Data Availability

The datasets generated during the current study are available from the corresponding author upon reasonable request.

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
