# Peer review of "Mono- and Bilayer Graphene/Silicon Photodetectors Based on Optical Microcavities Formed by Metallic and Double Silicon-on-Insulator Reflectors: A Theoretical Investigation"

_micromachines, 2023, doi:10.3390/mi14050906_

Round 1

Reviewer 1 Report

The manuscript presents a theoretical study investigating a photodetector, where the performance is improved by interference phenomena occurring inside an optical microcavity. However, there are numerous issues with English writing in the paper, and the structure is rather loose, making it challenging for readers to comprehend. Furthermore, the significance of the article is not immediately apparent, and the contribution of this study is not effectively highlighted. Therefore, the manuscript's novelty and scientific impact do not meet the standard for publication in Micromachines (MDPI).

Author Response

Point-by-point response to the reviewer's comments are in the uploaded file.

Reviewer 2 Report

The authors theoretically investigated the performance of graphene/silicon Schottky photodetectors based on Fabry-Pèrot optical microcavity optical microcavities formed by metallic and double silicon on insulator reflectors. They proposed using a metallic mirror to replace distributed Bragg mirror reported in the literature, which simplifies the processing technology of such device. Furthermore, the performance of the proposed device was compared with the state-of-the-art of similar devices. The overall quality of the paper is OK. However, before the paper can be accepted, some necessary comments should be addressed:  

1.     In section 2, the character explanations of formulas are confusing. For example, in descriptions of formula 1 (page 2), “q=1.602 x 10-19 C is the electron charge…”; in descriptions of formula 2 (page 3), the authors didn’t interpretate “?0” and “?s. The authors should check all formula descriptions in this paper.

2.     In table 1, only one previous work has been compared with the performance parameters of photodetectors in this work. The authors should provide more references for comparison so that readers could evaluate the device performance of this work.

3.     The authors used bilayer graphene to improve device performance due to its higher light absorption. So, does the device based on bilayer graphene have the best performance or does the device based on few-layer graphene have better performance? The authors should provide a theoretical explanation or predictions.

Author Response

Point-by-point response to the reviewer’s comments is in the uploaded file.

Reviewer 3 Report

The paper reports on reasonable results and can be published after minor revisions.

The authors’ main idea is to replace lossless but not efficient mirror based on a SOI structure by a gold film with higher reflectivity but also with higher losses. However, metal proximity reduces lifetimes of electrons (and holes) and may deteriorate responsivity but enhance response rate. This issue is to be discussed at least briefly (see, e.g. Gaponenko and Guzatov Proc IEEE, 2020).

Eq. 1 and relevant text.

What is absorption as a physical value? Which units are used for its measure? It should be explained. Photon energy and electron charge do not present in this equation but used in the text. These values are indirectly included in the 1.24 factor but this factor does possess dimension. This text should be carefully revised. What are the units to measure responsivity? If this is A/W (see, e.g. Gaponenko & Demir, Applied Nanophotonics, Cambridge 2019, p.386-387), then the right-hand part of Eq. 1 does not look like A/W value without explanations and comments. And the numerical factor 1242 should also be explained since usually it is taken to be 1239 (e.g., Gaponenko, Intro to Nanophotonics , Cambridge 2010, page 11.).

Eq. 3: Is it the original authors’ result or it is reproduced from the literature? Then reference is needed.

Why electron charge here is taken to be 1.58 instead of 1.60 x 10^-19 C?

Author Response

(The authors gave the same response as above.)

Round 2

Reviewer 1 Report

no more comment

Author Response

Thank you a lot for your help.